# Fibroinflammatory Liver Injuries as Preneoplastic Condition in Cholangiopathies

**DOI:** 10.3390/ijms19123875

**Published:** 2018-12-04

**Authors:** Stefania Cannito, Chiara Milani, Andrea Cappon, Maurizio Parola, Mario Strazzabosco, Massimiliano Cadamuro

**Affiliations:** 1Department of Clinical and Biological Sciences, Unit of Experimental Medicine and Clinical Pathology, University of Torino, Corso Raffaello 30, 10125 Torino, Italy; stefania.cannito@unito.it (S.C.); maurizio.parola@unito.it (M.P.); 2School of Medicine and Surgery, University of Milan-Bicocca, Via Cadore 48, 20900 Monza, Italy; chiara.milani@unimib.it (C.M.); mario.strazzabosco@yale.edu (M.S.); 3Department of Medicine (DIMED), Internal Medicine and Hepatology Unit, University of Padova, Via Giustiniani 2, 35121 Padova, Italy; cappon.andrea@gmail.com; 4International Center for Digestive Health (ICDH), University of Milan-Bicocca, Via Cadore 48, 20900 Monza, Italy; 5Liver Center and Section of Digestive Diseases, Department of Internal Medicine, Section of Digestive Diseases, Yale University School of Medicine, 333 Cedar Street, New Haven, CT 06520, USA; 6Department of Molecular Medicine (DMM), University of Padova, via Gabelli 63, 35121 Padova, Italy

**Keywords:** cholangiocytes, neoplastic transformation, cholangiocarcinoma, primary sclerosing cholangitis, Caroli’s disease

## Abstract

The cholangipathies are a class of liver diseases that specifically affects the biliary tree. These pathologies may have different etiologies (genetic, autoimmune, viral, or toxic) but all of them are characterized by a stark inflammatory infiltrate, increasing overtime, accompanied by an excess of periportal fibrosis. The cellular types that mount the regenerative/reparative hepatic response to the damage belong to different lineages, including cholagiocytes, mesenchymal and inflammatory cells, which dynamically interact with each other, exchanging different signals acting in autocrine and paracrine fashion. Those messengers may be proinflammatory cytokines and profibrotic chemokines (IL-1, and 6; CXCL1, 10 and 12, or MCP-1), morphogens (Notch, Hedgehog, and WNT/β-catenin signal pathways) and finally growth factors (VEGF, PDGF, and TGFβ, among others). In this review we will focus on the main molecular mechanisms mediating the establishment of a fibroinflammatory liver response that, if perpetuated, can lead not only to organ dysfunction but also to neoplastic transformation. Primary Sclerosing Cholangitis and Congenital Hepatic Fibrosis/Caroli’s disease, two chronic cholangiopathies, known to be prodrome of cholangiocarcinoma, for which several murine models are also available, were also used to further dissect the mechanisms of fibroinflammation leading to tumor development.

## 1. Introduction

Cholangiopathies are a large group of diseases of various etiology affecting the biliary epithelium, with a chronic-progressive trend and often invalidating outcome; although rare in prevalence if considered as a single entity, however, they are fairly common diseases, as a group. Cholangiopathies are characterized by a high morbidity and mortality together with a significant financial burden to health care systems and families. Cholangiopathies represent a consistent proportion in the indication for liver transplant (OLT) in the adult population, accounting for about the 16% in US in the period between 1998 and 2014, and are the main indication for the pediatric population, responsible for the 80% of the OLTx; in 2011 the costs for these interventions in the US reached $400 million [1,2]. In Italy, in 2008, diseases of the biliary tree were responsible for 10% of the hospitalizations for gastroenterological diseases (33,657 patients), and the average hospital stay was 6.5 days for a total of more than 220,000 days. The estimated annual social cost for the absence to the workplace account to about 890,000€ for each patient (Italian Association for the Study of the Liver, White Book, 2011. http://www.webaisf.org/pubblicazioni/documenti-finali-commissioni.aspx).

The etiology of cholangiopathies includes genetic and hereditary variants (Autosomal Dominant Polycystic Kidney Disease (ADPKD), Autosomal Recessive Polycystic Kidney Disease (ARPKD), Polycystic Liver Disease (PCLD), Chongenital Hepatic Fibrosis (CHF), Caroli’s disease (CD), and Alagille Syndrome (AGS)), immune-mediated diseases (Primary Biliary Cholangitis (PBC), Primary Sclerosing Colangitis (PSC), and Autoimmune Hepatitis (AIH)), pathologies associated to ischemia, infections, or toxic agents, such as drugs, alcohol, and herbal remedies, tumors (cholangiocarcinoma), and idiopathic diseases (Table 1) [3,4]. From a pathogenic point of view, cholangiopathies share numerous fundamental lesions including different degrees of portal inflammation, cholestasis, and the predisposition to develop biliary cirrhosis due to the progression of peribiliary fibrosis; moreover, some of them, namely PSC and fibropolycystic liver diseases (ARPKD, CHF, and CD), could predispose to cholangiocarcinoma (CCA) development [5,6].

The presence of an extensive portal inflammatory infiltrate coupled with intense peribiliary fibrosis is one of the most significant histopathological features for the most of the cholagiopathies, except for AGS, a genetic pathology due to mutation in *Notch2* or *Jagged1* genes and characterized by the near absence of biliary tree and sinusoidal fibrosis with scant inflammation. In fibroinflammatory diseases, bile ducts act not only as targets for the hepatic inflammatory response, for example in immune-mediated cholangiopathies such as PBC and PSC, but also promote the recruitment and activation of different cell types, among which are inflammatory cells and fibroblasts. 

## 2. Cholangiocytes as Pacemakers of Fibrosis Deposition

Biliary epithelial cells (BECs), also termed cholangiocytes, can recruit and activate inflammatory cells, thanks to their ability to secrete a range of pro-inflammatory cytokines and chemokines, which intervene in both the Th1 (i.e., interleukin (IL)-2, and interferon (IFN)γ) [7] and Th2 (i.e., IL-4, -5, and -6) [8] responses. Moreover, BECs have the ability to secrete growth factors, such as hepatocyte growth factor (HGF), platelet derived growth factor (PDGF)-BB, and -DD, connective tissue growth factor (CTGF), endothelin (ET)-1, vascular endothelial growth factor (VEGF), and transforming growth factor (TGF)-β2, able to stimulate the activation of mesenchymal cells through paracrine signals and therefore the production of extracellular matrix [4]. In fact, a histopathological finding characterizing the majority of the cholangiopathies is the deposition of fibrotic tissue in the portal space, in close proximity to reactive ductular biliary cells (RDCs), in accordance with the definition of ductular reaction, namely the hyperplasia of biliary structures resulting from biliary damage, as the “pacemaker” of liver fibrosis [9]. One of the most typical manifestations of cholangiopathies is the development of cholestasis, due to the effect of different mechanisms including an altered ductal secretion induced by the release of pro-inflammatory cytokines, as in immune-mediated cholangiopathies such as PSC and PBC, or an altered expression or functioning of the membrane transporters of BECs, induced by genetic defects, as occurs in cystic fibrosis, or finally the retention of hydrophobic bile acids that cause apoptosis death of hepatocytes and the subsequent development of hepatocanalicular cholestasis. 

## 3. Fibroinflammatory Response in Cholangiopathies

Fibrosis, as commonly accepted, is the consequence of excessive extracellular matrix (ECM) deposition during scar tissue formation after chronic liver injury. This process is associated with an increased and usually massive immune cells recruitment responsible for the modification of the hepatic scaffold and an altered angiogenetic activity, ultimately resulting in organ failure. In healthy liver, ECM normally present in the space of Disse, is composed by different Collagens (type I, III, IV, V, and VI), non-collagenous glycoproteins, such as laminin, and fibronectin, and proteoglycans [10]. In fibrotic liver, ECM deposition increases massively, up to tenfold, and also changes its normal composition, increasing the presence of collage I and III [10]. Fibrosis is associated with the formation of connective septa that interconnect the incoming vasculature (portal vein and hepatic artery branches) and outgoing vessels (central veins) [11].

The scar tissue disrupts the integrity of the sinusoid plexus, composed by permeable vascular spaces that are in direct contact with the hepatocytes, altering multiple metabolic functions of the liver. The shunting of blood through major fibrotic vessels bypasses the low pressure hepatic sinusoids causing portal hypertension, with increased risk of bleeding from esophageal or gastric varices, accumulation of ascites in the peritoneal cavity and immunodeficiency. Although fibrosis is no longer considered a static irreversible process in chronic diseases, the fibrotic process naturally leads to cirrhosis and to liver tumor formation. The balance of matrix-degrading activity, determined in turn by the balance of matrix metalloproteases (MMPs) and tissue inhibitors of MMPs (TIMPs), is important in both progressive fibrosis and liver fibrosis regression. Exacerbation of fibrosis resulting usually from chronic, non-resolving inflammation that triggers a wound-healing process that mitigates inflammatory tissue destruction but also leads to scar tissue formation [12].

Once the equilibrium is perturbed and liver fibrosis progress, immune system cells and in particular their mediators, play an active role in creating a fibrogenic milieu responsible for the tumor initiation in liver tissue and in particular, cholangiopathies, due to the fibroinflammatory microenvironment typical of these diseases are prone to develop neoplasias [13]. In this context, cholangiocytes are continuously stimulated by chemokines and cytokines, growth factors, and other soluble mediators that are responsible for damaging protoncogenes and tumor suppressor genes involved in cell growth, apoptosis, invasiveness and angiogenesis. 

In this review, we will elucidate the role of inflammation and of their mediators in the establishment of a fibroinflammatory response leading to the worsening of the chronic cholangiopathies and finally responsible for tumor initiation and development.

## 4. Cell Types Involved in Fibroinflammatory Response

### 4.1. Epithelial Cells

Hepatocytes and cholangiocytes, represent the two distinct liver epithelial cell types, which share several epithelial hallmarks, but differ radically in how they organize their apical and basolateral surface domains to fulfil different functions [14]. In cholangiopathies, BECs are the primary target of the pathogenetic sequence, but also contribute to disease development and recovery after damage [1]. Cholangiocytes line the biliary tree, a complex network of conduits inside and outside the liver [3], and regulate the fluidity and alkalinity of the bile secreted by the hepatocytes through a wide pool of transmembrane channels, transporters, and exchangers [15]. In addition, they actively perceive and respond to the inflammatory environment related to liver injury, as well as endogenous (bile acids, lipids, nucleotides) and exogenous (microbial derived, xenobiotic, and drugs) molecules that are able to modify cholangiocyte function and/or phenotype, affecting repair and remodeling of the biliary epithelium [16]. To restore bile duct integrity after damage, cholangiocytes must proliferate and then form branching tubular structures [17], a process originally termed “ductular reaction” [18]. These “reactive” cholangiocytes are characterized by motile properties and by the active production of growth factors (e.g., VEGF, PDGF, and TGF-β) [9,19,20] cytokines/chemokines (e.g., IL-6, IL-8, and tumor necrosis factor alpha (TNF-α)) [18,21,22], and morphogens (Hedgehog and Notch) [23], establishing an extensive cross-talk with other cell types of mesenchymal origin [17,24]. Moreover, liver “regeneration” can be sustained by the activation of the hepatic progenitor cells (HPCs), whose involvement depends on the severity and nature of the injury and on the residual replicative ability of hepatocytes and cholangiocytes [4]. HPCs are small cells, with an oval shape and scant cytoplasm, and the ability to differentiate into the biliary or the hepatocyte lineage [25]. HPCs are thought to reside at the interface of the terminal ductules and the canals of Hering [26], or at the peribiliary extrahepatic glands [27]. The differentiation of HPCs to the hepatocellular or biliary lineage depends on the type of inflammatory reaction developing around the stem cell niche and is finely regulated by fundamental morphogenetic signals such as Wnt/β-catenin, Hedgehog, and Notch [19,26,28,29]. While these mechanisms are necessary during cholangiopathies to repair the biliary damage, it is worth mentioning that they can also lead to undesired effects, such as the generation of an exaggerated neo-angiogenic response with production of new fibro-vascular stroma and progressive liver fibrosis [30]. Increasing evidence from murine models suggests that any type of liver epithelial cell, due to cellular plasticity, can give rise to cholangiocarcinoma (CCA), an aggressive malignancy of the biliary epithelium, characterized by late diagnosis and poor outcomes. In addition to the malignant transformation of cholangiocytes and adult hepatocytes, CCA may develop from HPCs localized in the canal of Hering or periductal glands stem niche [31] and HPC proliferation is directly related to cancer severity [32]. CCA deriving from HPCs presents phenotypic characteristics of both hepatocytes and cholangiocytes. Examples of CCA subtypes with HPCs origin are cholangiocellular carcinoma (CLC) and combined hepatocellular cholangiocarcinoma (CHC) [33,34,35]. Recently, Wu et al. [36] suggested that cell-cell and cell-matrix interactions are essential for HPCs function and CCA progression. They demonstrated that HPCs orchestrate the actions of myofibroblasts, immune cells, cytokines, matricellular proteins, and inflammatory proteins in HPCs niche through the expression of specialized CCN (cysteine-rich angiogenic protein 61 or CCN1; CTGF, or CCN2; nephroblastoma overexpressed or CCN3) proteins, in order to promote HPCs differentiation and CCA development. This finding opens new opportunities for future studies seeking to limit the uncontrolled activation of hepatic epithelial cells in order to prevent both cholangiopathies and CCA.

### 4.2. Mesenchymal Cells

Myofibroblasts (MFs) are specialized α-smooth muscle actin (α-SMA)-positive cell types involved in wound repair and liver fibrosis progression. In the liver there are multiple cell populations candidate to be myofibroblast precursors, including hepatic stellate cells (HSCs), portal fibroblasts (PFs) and fibrocytes. HSCs, constituting 5% to 15% of all hepatic cells, reside, in the normal liver, in the space of Disse, which is the area between hepatocytes and sinusoidal endothelial cells [37]. In normal liver HSCs, which are the major cellular store of Vitamin A and retinoid acid (RA) in the organism [38], display a quiescent phenotype expressing several markers including glial fibrillary acidic protein (GFAP), synaptophysin [39], desmin, and nerve growth factor (NGF) receptor p75 [40]. During chronic liver injury, quiescent HSCs gradually lose vitamin A and RA and differentiate in activated myofibroblast-like cells (HSC/MFs), acquiring proliferative, migratory, and contractile properties as well as starting to produce collagen Type-I; HSC/MFs are responsible for the deposition of the majority of ECM components [41,42,43] during fibrogenesis. Among HSC/MFs activation pathways, TGFβ and PDGF are considered the major fibrogenic mediators [43]. Literature data indicate that HSC/MFs, at least in murine models, are the predominant source of hepatic pro-fibrogenic MFs, regardless of the aetiology [44,45], although other cells, including PFs and bone marrow-derived mesenchymal cells, may represent additional sources of MFs. PFs are a population of resident fibroblasts, located in the portal tract mesenchyme that surrounds bile ducts, and expressing a specific marker profile including α-SMA, Collagen Type XV Alpha 1 (COL15A1), fibulin 2 (FBLN2), elastin (ELN), IL-6, cofilin 1 (CFL1), ecto-ATPase nucleoside triphosphate diphosphohydrolase-2 (NTPD2), and THY1 61 [43,46,47]. Moreover, similar to what was previously described for HSC/MFs, TGF-β represents the dominant profibrogenic cytokine involved in PFs activation [43,48]. The contribution of PFs to sustain fibrosis is considered particularly relevant in chronic cholangiopathies because PFs (i) act as “first responders” in biliary fibrosis [49]; (ii) maintain the structure of the intrahepatic bile ducts [50]; (iii) interact reciprocally with cholangiocytes, potentially influencing their state of activation and response to injury [43]. Lastly, less than 5% of the hepatic MFs may derive from bone marrow-derived cells including mesenchymal stem cells and fibrocytes. Fibrocytes, in particular, are mesenchymal cells arising from monocyte precursors that display both macrophages and fibroblasts features, co-express haematopoietic and progenitor cell markers (CD45 and CD34, respectively), and are able to produce ECM proteins. Fibrocytes have been reported to migrate into injured liver, possibly through signals operating on C-C chemokine receptor type 1 and 2 (CCR1 and CCR2) [51,52,53].

### 4.3. Immune Cells

Liver resident macrophages (Kuppfer cells, KCs) are responsible for the surveillance on liver homeostasis; typically, KCs can be found in both the centrilobular and periportal regions of the liver, and their functions and structures differ depending on their location. KCs embryologically derive from the yolk sac and, in normal conditions, represent up to 80% to 90% of all the macrophages of the entire human body [54]. Following liver injury, KCs massively replicate and are able to recruit monocytes from blood circulation, leading to an increase in local secretion of a bunch of peptides, among which inflammatory mediators (TNF-α, IL-1β), that activate HSCs through the Nuclear Factor Kappa-B (NF-κB) pathway, TGF-β and thrombospondin 1. HSCs activate MFs to secrete ECM proteins, IL-4, IL-13, and PDGF, which in turn stimulate collagen synthesis by MFs [55]. On the other hands, KCs are a reservoir of MMPs, in particular MMP12 and MMP13, produced in response to apoptotic cells phagocytosis, MMP9 and TNF-related apoptosis-inducing ligand (TRAIL), involved in the promotion of MFs apoptosis [55]. As previously discussed, macrophages could be considered a “double edge sword”, since they can mediate fibrosis/fibrolitic processes, an effect that could be explained by the heterogeneity of macrophages subset. In fact, the classical cataloguing M1, pro-inflammatory and pro-fibrotic subset, and M2, restorative and anti-fibrogenic, is nowadays considered simplistic and far from the real in vivo situation [56]. In murine models of chronic liver injury, KCs recruit from blood numerous different monocytes, in particular Ly6C^hi^ inflammatory cells, that once in the liver acquire the typical CD11b^+^/F4/80^+^ Kuppfer phenotype. These cells are responsible for massive release of pro-inflammatory cytokine, mainly IL-1β, TNF-α, IL-6, IL13, and TGF-β, which trigger HSCs and MFs toward fibrosis activation [11]. Once the ECM is deposed and scar tissue formed, another subset of immune cells, the LyC6^lo^ macrophages, can be activated, and originate by the switch of LY6C^hi^ due to the phagocytosis of apoptotic cells. This mechanism is responsible for high expression and secretion of MMPs and other anti-fibrogenic peptides, such as arginase 1 and chemokines receptor CX3CR1 in the local microenvironment [57]. In humans, similar heterogeneity is observed; in particular, in cirrhotic livers human KCs are characterized by CD14^lo^ and CD16^−^, while hepatic monocytes-derived macrophages as CD14^hi^, CD16^−^, CD14^+^ and CD16^+^ [55]. Neutrophils are the first defense line for the human body, as well as in the liver, but a clear role for this cell type in fibrotic/fibrolitic processes is still debated. Natural killer (NK) cells have an active role in anti-fibrotic mechanisms, mainly mediated by their expression of IFN-γ and of death receptors or ligands, such as Natural killer group 2-member D (NKG2D), TRAIL or Fas ligand (FasL), that modulate the activation of HSCs and MFs [55]. Moreover, NK cells are responsible for clearance of senescent HSCs, thus their deregulation may directly have affected the fibrotic/fibrolitic balance [58]. Furthermore, fibrotic livers are infiltrated by CD4^+^ T cells that actively participate to fibrosis progression. In particular, Th2 cells increase fibrosis deposition by secreting IL-13 and leading to the hyper-expression of TGF-β1 by HSCs, ultimately resulting in MMP9 expression and secretion, which, in turn, cleaves more pro-TGF, to further sustain fibrogenesis [11]. On the other side, IFN-γ released by Th1 CD^+^ cells alters the MMPs/TIMPs balance towards MMPs production, and activates fibrolisis. Another subset of T cells likely actively involved in fibrosis progression are the Th17, that release IL-17, which activates its cognate receptor expressed on HSCs surface, to stimulate the signal transducer and activator of transcription (STAT)3-dependent synthesis of collagen I [13]. Tregs cells are putatively the counterpart of Th17 because this cell type secretes IL-10 that, through the phosphorylation of STAT5, contributes to ameliorate fibrosis in rodent models of BDL and parasitic infestation [10].

## 5. Epithelial-Mesenchymal Cross-talk

In cholangiopathies, to restore the normal liver architecture, cell types involved in the reparative/regenerative hepatic response, i.e., BECs (RDCs, and HPCs), cells of mesenchymal origin (PFs, and HSCs), and inflammatory infiltrate (macrophages and neutrophils) interact each other exchanging numerous peptides and sharing several ligands and receptors [17,59] (Figure 1). If this finely tuned response is incorrect, this could lead to the establishment of a chronic cholangiopathy, accompanied by the deposition of fibrosis, eventually leading to cirrhosis, and by a massive hepatic inflammation. The interactions among these cell milieus are mediated by three different categories of peptides that could be grouped as growth factors, proinflammatory and profibrotic cyto/chemokines, and morphogens. 

Here, we briefly describe the functions of the main mediators involved in this cross-talk. 

### 5.1. Growth Factors

The most studied and known growth factor involved in hepatic fibrogenesis is TGFβ. Once liver insult is received, several cell types, among which cholangiocytes, MFs, HSCs, and KCs start to actively secrete TGFβ, in particular the isoforms β1 and β2 [60]; this growth factor stimulates its receptors TGFβRI and RII to activate a profibrogenic response through the activation of the Smad2/3-mediated signaling [61]. Once phosphorylated, Smad2/3 create a multicomplex with Smad4 that migrates to the nucleus to exert its transcriptional activity leading to the transcription of target genes, among which collagens (I and IV), α-SMA, and MMPs [62]. Notably, TGFβ ligands are secreted mainly by HSCs, endothelial cells (ECs) and KCs, that, by an autocrine-paracrine loop, are responsible for the proliferation, activation and secretion of ECM proteins by HSCs. Despite the fact that cholangiocytes are responsible, only in a minimal part, for the secretion of this growth factor, in cholangiopathies the presence of RDCs is fundamental for the activation of the latent TGFβ. In chronic cholangiopaties, in fact, RDCs aberrantly express de novo αVβ6 integrin [63,64] that, together with its function as a mediator of the cell-matrix interaction, is necessary for the modification of the so called “Small Latent Complex”, composed by the latent form of TGFβ interacting with Latency Associated Peptide (LAP) [65]. αVβ6 integrin cleaves the TGFβ/LAP complex and, once TGFβ is released as a monomer, it can dimerize thus acquiring the ability to bind its specific receptor TGFβR [65]. VEGF and receptors are a family of peptides composed of six ligands, VEGF-A to -E and by placental growth factor, that bind their specific receptors VEGFR1, R2 and R3. Despite the initial belief that they exert their proliferative effects on ECs only, mounting data demonstrate that VEGF ligands could be secreted by several cell milieu and could activate cellular responses on cholangiocytes [66,67], HSCs [68], and macrophages [69]. In ADPKD, the proliferation of liver cysts both in human and in animal models of PC2 conditional KO mice, VEGF-A is hyper-secreted by BECs and stimulates liver cyst enlargement in an autocrine fashion through a VEGFR2/(Mitogen-Activated Protein Kinase Kinase 1) MEK/(Extracellular Signal-Regulated Kinase) ERK1/2/(Mammalian Target Of Rapamycin) mTor axis [66,70,71]. Similarly, in the bile duct ligated (BDL) mouse model of chronic liver insult, VEGF secreted by RDCs sustained the expansion of both RDC and HPC compartments [72]. VEGF secretion could exert chemotactic and proliferative response in HSCs and also fuel their collagen secretion in the liver microenvironment, thus contributing to the development of portal fibrosis and to the worsening of the disease [73]. One of the most relevant family of growth factors involved in the establishment and development of portal fibrosis is the Platelet derived growth factor (PDGF). PDGF family is composed by five ligands (four homodimers PDGF-AA, -BB, -CC, and -DD, and one heterodimer PDGF-AB) and by three cognate receptors, PDGFRα, PDGFRβ, and PDGFRαβ [74]. PDGF signaling supervises several physiological mechanisms, both in normal and diseased conditions. In particular, the secretion of PDGF-B by RDCs of BDL mice stimulates the proliferation and collagen secretion of HSCs; moreover, PDGF-B induces the chemotactic recruitment of mesenchymal cells and the transformation and activation of PFs to MFs, which are stimulated to proliferate and to actively produce ECM components through the activation of the (Phosphatidylinositol-4,5-Bisphosphate 3-Kinase) PI3K signaling [75]. Another PDGF ligand reaching even more importance is PDGF-D that in CCA, once secreted by neoplastic cholangiocytes, is responsible for the recruitment of cancer associated fibroblasts (CAFs) by the concomitant activation of (C-Jun N-Terminal Kinase) JNK signaling and Rho GTPases Rac1 and Cdc42 [76]. A profibrotic growth factor secreted in chronic liver diseases by a vast number of cells, such as cholangiocytes, PFs, MFs, HSCs, and mast cells, is connective tissue growth factor (CTGF, or CNN2). One of the factors involved in CTGF upregulation is TGFβ, that upregulates its expression and secretion by HSCs mainly through (Activin Receptor-Like Kinase) ALK5/JAK1/Stat3 [77] and ERK1/2 signaling [78]. Furthermore, CTGF has a potent chemotactic and trophic effect on HSCs and induces their collagen secretion [79]. Chronic cholangiopathies characterized by an intense ductular reaction and fibrosis deposition, such as biliary atresia and PSC, are characterized by a strong secretion of CTGF [80,81] thus implying a potential pathogenic role of this growth factor in the worsening of these diseases.

### 5.2. Morphogens

Different morphogenetic programs are deregulated in the onset and development of liver diseases; here, we briefly cast an eye on the main signal pathways involved in the pathogenesis of fibroinflammatory cholangiopathies. For instance, in normal conditions β-catenin acts as a structural protein of the adherent junctions being a bridge between E-cadherin at the cell surface and α-catenin and f-actin [82]. Usually β-catenin is maintained in an inert state into the cytoplasm by a so called “destruction complex” composed by several proteins that stabilize it. In cholangiopathies, β-catenin is phosphorylated and, if not ubiquitinated and destroyed, could reach the nucleus to activate the WNT/β-catenin signaling. In the nucleus, β-catenin binds its specific transcription factor T-cell factor/lymphoid enhancer factor (TCF/LEF). This mechanism is responsible for the transcription of several peptides, among which fibroinflammatory cyto and chemokines, such as chemokine (C-X-C motif) ligands (CXCL)1, 10, and 12, and CTGF [64,83]; β-catenin nuclear entry represent a fundamental step for the initiation of the proliferation and enlargement of RDC plexus [84] and is a key regulator of the proliferation of HPC in adults suffering for different chronic liver diseases [85]. Furthermore, the activation of this signaling is involved in HSC-MF transdifferentiation and activation, and in BDL mice, the inactivation of this signal pathway using an adenovirus carrying Dickkopf-1, a WNT antagonist, is able to significantly reduce the extent of fibrosis [86]. Moreover, its aberrant phosphorylation at the Ser675 of β-catenin in a mouse model of CHF/CD, act as protein stabilizer, impeding its phosphorylation and degradation [87] thus stimulating the secretion of profibrotic and proinflammatory mediators by cystic epithelia [64]. Another fundamental pathway involved in RDC proliferation is Notch [88], a signal mechanism requiring the cell-cell contact and particularly important in biliary differentiation and elongation along bile duct fetal development [28,89]; Notch (1 to 4) is a family of receptors for different ligands (Jagged, and Delta) that, once docked, allow the cleavage by γ-secretases of the notch internal cellular domain (NICD), that translocate to the nucleus to act as the transcription factor binding the recombinant signal binding protein for immunoglobulin kappa J (RBP-Jk) [17]. Notably, mutations of Notch2 or of its ligand Jagged1 are responsible for the development of Alagille syndrome, a recessive genetic disease characterized by a lack of biliary structures that fail to elongate from the hilum [89] and characterized by an accumulation of cells of intermediate hepato-biliary morphology [90]. Recent papers outlined the importance also in the activation of HPC niche in liver disease, and in the elongation of RDC as a fundamental inducer of the hepatic reparative/regenerative response to liver injury [19,91]. Moreover, Notch signaling is involved in inflammatory responses, acting concomitantly with TLR4, to stimulate the secretion of IFNγ, and of chemokines, among which CXCL10 that is able to recruit infiltrating macrophages and to stimulate their M1 polarization [92]. A third fundamental morphogenic signaling involved in the pathogenesis of cholangiopathies is Hedgehog (Hh). Hh deregulation could act at a different level in hepatic regenerative mechanisms responsible for fibrosis deposition and inflammation. During liver fetal development it stimulates the proliferation of hepatoblasts [93] and maturation of fetal biliary epithelia by controlling the expression of (Yes Associated Protein 1) YAP and Sox9, both transcription factors necessary for biliary specification [94]. Notably, in liver diseases such as PBC [95], MFs, and RDCs express high levels of Hh that could act in an autocrine/paracrine manner on surrounding cells and MFs itself further stimulating the proliferation of BECs, through the upregulation of PDGF-B signaling in an (Protein Kinase B Alpha) Akt-dependent fashion [96].

### 5.3. Proinflammatory Cyto and Chemokines

Another group of peptides deeply involved in the cross-talk among the different hepatic cell types are the prionflammatory and profibrotic cytokines and chemokines. The diseased liver microenvironment is usually full of such mediators that regulate the different pathways activated during liver regeneration, among which Interleukins (IL-1, -6, -8, -13, and -33), monocyte chemotactic protein (MCP)-1, IFNγ, and chemokine (C-X-C motif) ligands (CXCL1, 10, and 12). One of the most important cytokines involved in liver regeneration is IL-6, the founder member of a large family of peptides primarily involved in cell proliferation. It could be secreted by several cell types, such as HSCs, KCs, and cholangiocytes, and, interacting with its receptor IL6R, it heterodimerizes with the co-receptor gp130, leading to epithelial cells proliferation following phosphorylation of Janus kinase (JAK) and STAT3 [97]. Other fundamental groups of ILs in the pathogenesis of chronic cholangiopathies, in particular of biliary atresia, are IL-13 and IL-33. Following RRV infection, in fact, RDCs start to secrete IL-33, which primes innate lymphoid cells type 2 (ILC2) to secrete a wide range of ILs and growth factors, such as IL-13, Osteopontin, and TGFβ. IL-13, in turn, further stimulates the proliferation of RDCs through a paracrine loop, sustaining the reactive ductular response typical of this disease [98]. MCP-1 is a crucial chemokine responsible for the recruitment of monocytes at the site of inflammation [99]. MCP-1 released by inflamed cholangiocytes interacts with its receptor CCR2 expressed by a subclass of circulating monocytes (CCR2^+^/CD14^+^/CD16^−^) that are actively recruited and once transdifferentiated to macrophages are likely mainly responsible for the activation of HSCs at the site of damage [100]. Moreover, MCP-1 is known to stimulate PFs proliferation, transdifferentiation to MFs and the transcription of procollagen-1 precursors [101]. Fibroinflammatory response in cholangiopathies is driven also by members of the CXCL family; the most studied is CXCL12, also known as SDF-1, a chemokine with a prominent chemotactic on inflammatory cells, in particular monocytes, lymphocytes, hematopoietic stem cells, and B cell precursors. PBC and PSC are characterized by BECs hyperexpression of CXCL12, responsible for the recruitment of T cells carrying its specific receptor C-X-C chemokine receptor type (CXCR) 4, the cognate receptor of CXCL12, and sustaining the long-standing inflammatory state of these immune mediated diseases [102]; moreover, HSCs are also responsive to CXCL12, that excites their proliferation and ability to secrete collagen [103]. Biliary cysts of CHF are characterized not only by the aberrant secretion of CXCL12, but also of other cytokines, in particular CXCL1 (or KC), and CXCL10 (or IP-10) [64]. CXCL1 is a chemokine secreted by numerous cell types, including epithelial cells, HSCs, macrophages, and neutrophils. It is mainly involved in chemoattraction and proliferation of neutrophils [104], but several papers demonstrate that its effect is not limited to this inflammatory cell type; CXCL1 could, in fact, stimulate proliferation of HSCs in an autocrine loop, and inhibition of the PI3K/Akt pathway is able to slow cell proliferation and, in vivo, to reduce development of fibrosis in a mouse model of carbon tetrachloride (CCl_4_) intoxication [105]. Moreover, stimulation of TLR4 increases NF-kB-mediated secretion of CXCL1 in a CFLD mouse model, likely responsible for the accumulation of neutrophils in CFTR-related liver diseases [106]. CXCL10, through its receptor CXCR3, is involved in several pathophysiologic responses, in primis recruitment of inflammatory cells, such as monocyte and macrophages, and NK and T cells, and also in angiogenesis, cell adhesion and fibrogenesis. CXCL10 is upregulated in response not only to IFNγ [107], but also by the activation of the β-catenin signaling [64]. Hypersecretion of CXCL10 in FPC-KO mice is responsible for the pericystic accumulation of macrophages and their M1-M2 transformation [64]. Moreover, the inhibition of its receptor with AMG-487, in vivo, leads to a reduced fibrosis deposition accompanied by reduction in cystic area [83]. Furthermore, CXCL10 is considered an independent biomarker of fibrotic development in HCV-transplanted patients [108] and correlates with apoptosis in HCV patients [109]. It is worthy to highlight that secretion of inflammatory mediators is involved not only in the recruitment and activation of inflammatory cells and fibroblasts, but that their accumulation could lead to the persistent establishment of a chronic phlogistic response and in the hyper activation of enzymes directly involved in the neoplastic degeneration of epithelial cells to originate cholangiocarcinoma. 

## 6. Mechanisms of Neoplastic Transformation

Cholangiocarcinoma (CCA) is a relatively rare and highly aggressive malignancy originating from the neoplastic transformation of the intra or extrahepatic biliary tract epithelia. The incidence of CCAs shows a marked variation worldwide, with intrahepatic CCAs being influenced by racial and gender differences (CCA incidence is slightly higher in men) which is not significant in the case of extrahepatic CCAs [110]. Although the aetiology of most CCAs remains undetermined [111] and risk factors may vary geographically [110], several studies have identified a close relationship between CCA development and chronic biliary tract infection and inflammation and, in particular, with the pre-existence of chronic fibroinflammatory liver diseases, such as PSC and CD. Moreover, it is well established that the oxidative stress and DNA lesions, as consequences of long-lasting inflammatory processes, as in chronic cholangiopathies, play a key role in CCA development (Figure 2). In particular, the persistent portal inflammation results in a sustained generation of endogenous compounds, such as reactive oxygen species (ROS) and inducible nitroc oxide synthase (iNOS), that activate several pro-inflammatory cytokines including TNF-α, IFNγ, TGFβ, and different ILs, thus causing accumulation of nitric oxide (NO) in cholangiocytes, leading to nitrosative stress [112,113], finally enhancing CCA progression. Oxidative and nitrosative stress can induce malignant transformation of epithelial cells directly, by inducing oxidative DNA lesions, characterized by multiple DNA single or double-strand breaks and genetic instability, as also described for other cancers, or by inhibiting DNA repair enzymes, as well as, indirectly, by affecting the cell proliferation/apoptosis ratio [113,114,115,116]. 

In particular, nitric oxide (NO) and peroxynitrites are responsible for the formation of 8-oxo-7,8-dihydro-2′-deoxyguanosine (8-oxodG), and 8-Nitro guanine, two mutagenic compounds able to inhibit DNA base repair and cause G → T transversions. More relevantly, a study from Thanan and colleagues [117] demonstrated that high levels of 8-oxodG was observed in the urine and leukocytes of Opisthorchis Viverrini-infected patients also presenting CCA, thus suggesting the use of urinary 8-oxodG as biomarker to monitor not only infestation but also carcinogenesis. NO and peroxynitrites may also favor the progressive accumulation of DNA damage in CCA by inhibiting the activity of DNA damage reparative systems, as well as through the inactivation of 8-oxo-deoxiguanine DNA glycosylase 1 (hOGG1, the most important DNA repairing enzyme) [113,118]. Thanan and collaborators [111] also hypothesized that CCA could derive from the neoplastic transformation and persistence of liver stem/progenitor cells stimulated by the condition of chronic and inflammatory liver injury. In particular, they observed a significant correlation between increased levels of 8-oxodG and CD133- and/or Oct3/4-positive cells compared to those in negative tissues [111]. Enhanced proliferative signaling and evasion of apoptosis may constitute additional steps for the development of CCAs. Concerning the effects of nitrosative and oxidative stress on the regulation of the balance between epithelial cell proliferation and apoptosis, in conditions of persistent inflammation, accumulation of NO and its derivatives are supposed to be responsible for the increased cell proliferation as well as for the inhibition of pro-apoptotic mechanisms.

Furthermore, the iNOS-dependent COX-2 activation through the p38 MAPK/JNK axis, leads, in CCA, to the generation of high amounts of prostaglandin E2 (PGE_2_), which in turn, by binding its specific receptor EP1, can induce a tumorigenic effect by enhancing neoplastic cell proliferation and invasiveness. In addition, PGE_2_ is able to transactivate the epidermal growth factor receptor (EGFR), through a Src binding protein-mediated mechanism, resulting in the activation of PI3K/AKT signaling pathway, known to be responsible for proliferative and anti-apoptotic responses in CCA cells [119,120]. In particular, the activation of the PI3K/AKT pathway inhibits the Bad-Bcl_xl_ binding [121] and the consequent mitochondrial Bax translocation, a critical step for activating the caspase cascade. This anti-apoptotic effect induced by EGFR activation is counteracted by the treatment with celecoxib, a selective COX-2 inhibitor [122]. On the other hand, NO can directly inhibit the activation of caspases 3, 8, and 9, through nitrosilation of the cysteine residues present in the active site of these caspases [123]. Neoplastic transformation could be also stimulated by xenobiotics that could act as carcinogens, inducing directly inflammatory damage responsible for genetic alterations or co-carcinogens, namely, substances, usually non-cancerogenetic, that act in concert with carcinogens in promoting the onset of tumors. A known pro-neoplastic stimulus is the chronic administration of thioacetamide (TAA), that induces liver injury followed by cirrhosis and cholangiocarcinoma development after 30 weeks in rats [124]. Notably, the accumulation of bile acids following bile duct ligation could burst the neoplastic development induced by TAA-treatment in rats by generating a more sustained inflammatory environment and inducing RDCs proliferation respect to the TAA-treatment only [125].

Recent data have also highlighted a critical role for Notch signaling in the formation of biliary tract cancers [126] as well as an enhanced Notch expression in PSC patients and in those carrying CCAs. In particular, NO-dependent activation of iNOS may also promote antiapoptotic mechanisms by interfering with Notch signaling. Ishimura and colleagues [127] showed that in mouse CCA cell lines, NO is able to induce the cleavage of the NICD1, which translocates into the nucleus to form a transcriptional complex with recombination signal-binding protein 1 for J-kappa (*RBPj*), up-regulating several target genes, among which hairy and enhancer of split-1 (*Hes-1*). This mechanism, mediated by the activation of the JNK1/2 signaling, is believed to be responsible for increased cell survival and reduced apoptosis of CCA cells [127].

## 7. Fibroinflammatory Cholangiopathies Heralding CCA

As previously described, some chronic fibroinflammatory biliary diseases could lead to the development of CCA. The best known and studied nosologic conditions prodromal to neoplastic transformation are PSC, and CD (Table 1); here, we briefly describe the main features of these two diseases, the molecular mechanisms postulated to be involved in their tumoral degeneration, and the main mouse models available for their study.

### 7.1. Primary Sclerosing Cholangitis (PSC)

PSC is a rare chronic cholangiopathy of autoimmune origin that affects about 250,000 individuals in the EU and 200,000 people in the US. PSC is characterized by a strong peribiliary fibrosis and inflammation, which can affect bile ducts of any size, from the smallest branches of the intrahepatic ducts, to the extrahepatic major ducts. Its incidence in Western countries varies from 0.2% to 20% per year, with higher percentage in the Scandinavian population, and a mild prevalence (60% to 65%) in men; PSC often affects the young adult population (30–40 years), and is accompanied, in 70% of cases, by chronic intestinal inflammatory diseases (IBD), in particular ulcerative colitis [128]. Histologically, its pathognomonic lesion is the development of a thick concentric fibrotic hose around the biliary tree leading to the development of cholangitis with fibro-obliterative evolution (onion-skin like fibrosis), which determines the progressive vanishing of the bile ducts. The inflammatory infiltrate contains mainly T lymphocytes and natural killer cells, which, through the secretion of IL-21, exert a potent effect on fibroblasts proliferation and activation [129]. The prognosis of PSC is very variable, but not infrequently the natural history of the disease is burdened by the high risk of developing malignant tumors, not only at the biliary (cholangiocarcinoma), but also at the intestinal (colorectal carcinoma) level. Unlike other cholangiopathies, such as PBC, the effectiveness of ursodeoxycholic acid (UDCA) is not well recognized, and the only curative option remains the OLT, with a risk of recurrence in 30% of cases [130]. 

Different animal models for PSC are currently available for research purposes [131,132]. For example, Mdr2 knockout (Mdr2^−/−^) mice were obtained by genetically disrupting the *Mdr2* (*Abcb4*) gene, which is a mouse orthologue of human *MDR3* (*ABCB4*), encoding for a canalicular flippase expressed by hepatocytes, mediating the transport of biliary phospholipids into the outer leaflet of the canalicular cell membrane, which enables their subsequent secretion into the bile. Inactivation of the Mdr2 gene in mice produces periductular onion-skin type fibrotic lesions and pronounced ductular reaction, liver fibrosis, early-onset severe portal hypertension, and increased transcripts of Col1a1. In addition, *Mdr2^−/−^* mice present clinically relevant cirrhosis complications, such as primary liver cancers [133]. Another genetically induced PSC-like model is the cystic fibrosis transmembrane conductance regulator knockout (*Cftr^−/−^*) mouse, which harbors a mutation of exon 10 of the *Cftr* gene. *Cftr^−/−^* mice develop focal cholangitis with thickened bile and bile duct proliferation, finally resulting in biliary cirrhosis [134].

Regarding the association of PSC with the development of CCA, PSC is responsible for 10% of the diagnosis for biliary neoplasia [135], in particular for young adults; the annual risk of developing this neoplasia for patients suffering for PSC ranges from 0.6% to 1.5% and up to the 50% of the diagnoses for PSC are followed by CCA development in the first 2.5 years [136,137,138]. Since PSC affects all the districts of the intra and extrahepatic biliary network, CCA development could be associated with the entire course of the biliary tree and often is proceeded by preneoplastic lesions such as intraepithelial biliary or intraductal papillary neoplasias, but PSC patients usually develop mucinous or infiltrating desmoplasic CCA [139,140]. The mechanisms leading to the neoplastic transformation of PSC to CCA are actually unclear, but are associated with the persistence of a chronic inflammatory state of the biliary tree. It was proposed that, in PSC, neoplastic transformation was associated to mutations of the tumor suppressor p53, and of the oncogene KRAS [141,142]. Furthermore, recent studies outlined the impact of NO production by iNOS as a stimulator of neoplastic transformation. NO in fact, not only stimulates the expression of NOTCH, a signaling known to be involved in cholangiocarcinogenesis, but can directly induce nicks in dsDNA [118], hamper the activity of enzymes involved in DNA proofreading and repair [115], and impair the expression of oncosuppressors, such as p16 [143].

### 7.2. Caroli’s Disease (CD)

CD is a genetic cholangiopathy caused by mutations in polycystic kidney and hepatic disease 1 (*PKHD1*), the gene coding for fibrocystin/polyductin (FPC), a large protein with a single transmembrane domain expressed by cilia and centromers of cholangiocytes and renal epithelial cells [144]. The FPC role remains unknown, but is thought to be involved in different cellular functions, including regulation of cell proliferation, differentiation, secretion, tubulogenesis, and cell-matrix interaction [145]. Due to FPC deficiency, CD is characterized by cystic dysgenesia of the intrahepatic bile ducts, which preserve an immature phenotype (ductal plate malformations) and by portal fibrosis, resulting in clinically relevant portal hypertension and related complications [3,17,146]. The mechanisms responsible for portal fibrosis in CD are still unclear, but recent studies have shown that FPC-defective cells present several modifications in signaling mechanisms, including Ca^2+^ homeostasis [147], 3′–5′-cyclic adenosine monophosphate (cAMP) [148] and mTOR [149] pathways. In FPC-defective cholangiocytes, the increased cAMP/PKA signaling, which results in the stimulation of cell proliferation and cyst expansion [148], is accompanied by PKA-dependent β-catenin activation [150]. Indeed, PKA is able to phosphorylate β-catenin at Ser-552 and Ser-675, different from those classically phosphorylated by Casein Kinase 1 Alpha 1 (CK1) and Glycogen Synthase Kinase 3 Beta (GSK3), preventing β-catenin degradation and resulting in its transcriptional activity [151]. Notably, recent findings suggest that the aberrant activation of β-catenin in FPC-defective epithelial cells causes a chronic, low-grade inflammatory response [64] named “parainflammation”, a process of adaptation to an unresolving cell dysfunction or noxious conditions [152]. When cell dysfunction is persistent, the inflammatory response, unable to restore the normal tissue homeostasis, becomes pathologic and can ultimately lead to scarring [152,153]. Thus, the development of fibrosis in CD depends on a complex interplay between epithelial and inflammatory cells, and exploring the factors and the mechanisms that take part in this relationship may help in discovering new therapies against cholangiopathies. This goal could be achieved by means of the *Pkhd1^del4/del4^* mouse, a well-established model of CD obtained with a targeting construct designed to inactivate Pkhd1 gene by disrupting exon 4 in mice with a C57BL6/129 mixed background. *Pkhd1^del4/del4^* mice are characterized by intrahepatic bile duct proliferation, progressive cyst formation, periportal fibrosis, and extrahepatic manifestations, including pancreatic cysts, splenomegaly, and common bile duct dilation [154]. Another important model is the *Pkhd1^del2/del2^* mouse, obtained deleting Exon 2 of the Pkhd1 gene and replacing it with a neomycin resistance cassette flanked by loxP sites, which could be subsequently removed by Cre-lox recombinase. Phenotypically, the *Pkhd1^del2/del2^* mouse is similar to the *Pkhd1^del4/del4^* mouse, developing altered bile ducts and enlarged portal triads, with portal fibrosis and cystogenesis [155]. Finally, the polycystic kidney (PCK) rat has been suggested as a useful and promising animal model to study CD. The PCK rat is a spontaneous mutant derived from a colony of Crj:CD rats, and was found to show multiple segmental and saccular dilatations of the intrahepatic bile ducts, ductal plate malformation, hepatic fibrosis and gross enlargement of kidney and liver [156].

CD is also considered a risk factor for CCA development, increasing the risk for cholangiocarcinogensis by 30-fold [135]. In CD patients, and generally in patients with bile duct cystic disorders, CCA is typically diagnosed at a mean age of 32 years, with lifetime incidence ranging from 6% to 30% [157]. As for others risk factors, the malignant transformation of cholangiocyte in CD conditions arises in a background of chronic inflammation, in which the high amount of cytokines and factors secreted as a result of the aberrant activation of β-catenin in FPC-defective cholangiocytes, triggers and maintains the process of cholangiocarcinogenesis [158]. Molecules participating in this process promote neoplastic transformation by altering protooncogenes, DNA mismatch repair genes or proteins, and tumor suppressor genes involved in cell growth, apoptosis, invasiveness, and neoangiogenesis, inducing uncontrolled cell proliferation and invasion [159]. Therefore, reducing or regulating the complex interplay between epithelial and inflammatory cells in CD patients could help in monitoring not only the disease but also side effects, such as CCA development.

## 8. Conclusions

Most of the cholangiopathies share several pathological features, such as the peribiliary accumulation of different cell types, among which fibroblasts and macrophages, whose intense cross-talk is responsible for the initiation and persistence of the phlogistic processes and of fibrosis deposition that accompanies these pathologies. Alongside these manifestations, cytokines, chemokines, and growth factors released by the different cell populations, lead to the generation and hyper-expression of reactive oxygen and nitrogen species that can cause damage to the genetic heritage of the biliary cells that can predispose to the development of CCA. However, currently, research is limited by the fact that, despite the existence of many well-established animal models of cholangiopathies and of CCA, the investigators are still lacking animal models of neoplastic transformation arising from cholangiopathies, useful for following the different steps of CCA development, from the establishment of chronic inflammatory state, to the formation of dysplastic nodules and then progressing to frank CCA. Thus, the above described rodent models of fibropolycystic liver disease (*Pkhd1^del2/del2^*, *Pkhd1^del4/del4^*, and PCK,) or PSC (*Mdr2^−/−^*, and *Cftr^−/−^*), could be successfully employed to evaluate if their mutations enable CCA carcinogenesis after toxic insults, for example following chronic oral administration of TAA. Unravelling the molecular mechanisms at the basis of fibroinflammatory responses in chronic cholangiopathies and the routes leading to neoplastic transformation of the biliary epithelia, is a key step for the development of new strategies to closely follow patients to prevent neoplasia development; furthermore, advances in the knowledge of intimate molecular pathogenesis of cholangiopaties could lead to new therapeutic approaches, actually lacking. 

## Figures and Tables

**Figure 1 ijms-19-03875-f001:**
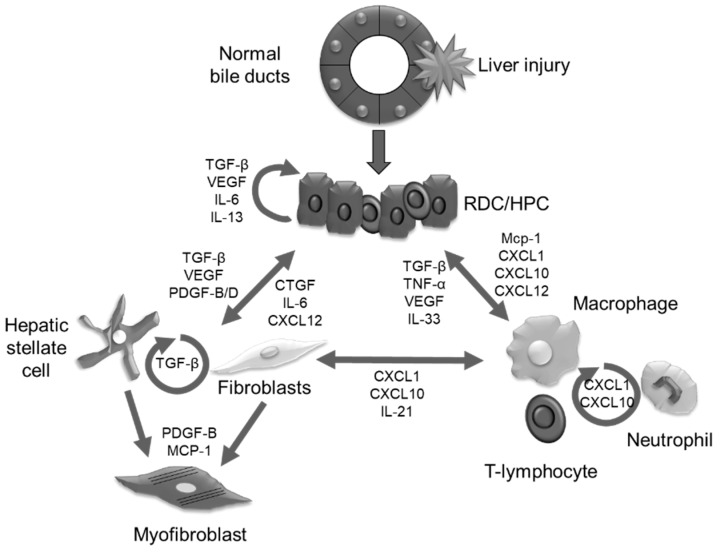
Autocrine and paracrine signaling characterizing the cross-talk among the different cell types involved in the development of cholangiopathies. Following a bile duct insult, cholangiocytes start to secrete several mediators involved in the recruitment and activation of mesenchymal, as well as inflammatory cells. Under the continuous stimulus induced by the chronic damage, fibroblasts could accumulate in the portal tract and, together with hepatic stellate cells, could transdifferentiate to myofibroblasts, directly responsible for the accumulation of periportal fibrosis. Similarly, damaged bile ducts could recruit different types of inflammatory cells, among which T lymphocytes, neutrophils, and macrophages (both M1 and M2), that further sustain the development of the disease.

**Figure 2 ijms-19-03875-f002:**
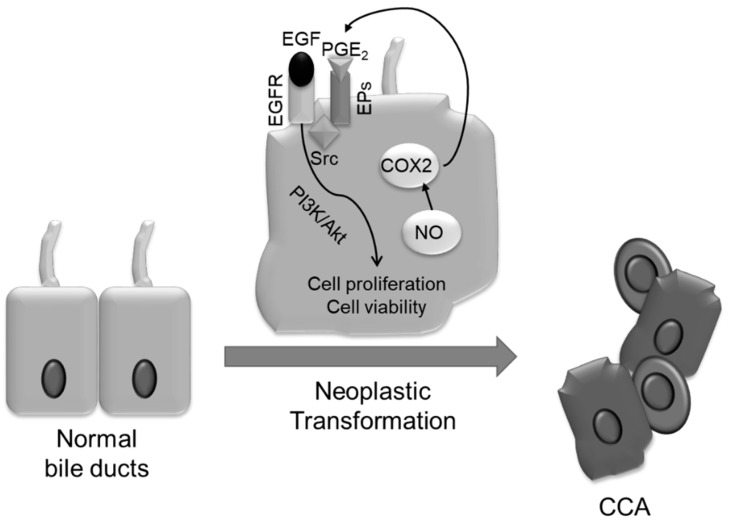
Chronic inflammation is responsible for the neoplastic transformation of the biliary epithelial cells. Alterations are due to the presence in the hepatic microenvironment of several molecules, such as peroxinitrites and oxygen free radicals that could induce the accumulation of DNA damage. Nitric oxide (NO) is able to stimulate cell proliferation and escape from apoptosis through the activation of COX2 that stimulates the p38 MAPK/JNK axis to secrete PGE_2_, that further transactivates the epidermal growth factor receptor (EGFR); this latter receptor also activates the PI3K/AKT pathway responsible for the proliferation and resistance to apoptosis in cholangiocarcinoma (CCA).

**Table 1 ijms-19-03875-t001:** List of cholangiopathies of different etiology characterized by fibroinflammation.

Cholangiopathies	Incidence	Mutations	Pathogenesis	Clinical Features	CCA Development
ADPKD	1:400–1000	85% to 90% PKD1 (polycystin 1), 10% to 15% PKD2 (polycystin 2)	Ductal plate malformation, biliary microhamartomas and cysts, with scant fibrosis and inflammation	Renal, hepatic and pancreatic cysts, renal failure; complications: Rupture, infections and haemorrhages, often associated with cerebral aneurysm (20% of cases) and cardiac valves abnormalities	Rare
ADPLD	1:100,000	PRKCSH (hepatocystin) and SEC63 (endoplasmic reticulum translocator)	Ductal plate malformation, biliary microhamartomas and cysts	Similar to ADPKD except for renal implications	Rare
Alagille Syndrome	1:30,000	50% to 60% JAG1 and rarely NOTCH2	Progressive vanishing of intrahepatic bile ducts	Biliary ductopenia, conjugated hyperbilirubinemia, and liver failure. Extrahepatic manifestations may involve heart, kidneys, skeleton and face	Unknown
ARPKD	1:20,000–40,000	PKHD1 (fibrocystin/polyductin)	Ductal plate malformation, biliary microhamartomas, and cysts; peribiliary fibrosis and inflammation	Recurrent cholangitis, hepatic cysts and microhamartomas, and portal hypertension. Renal multiple cysts, and kidney failure	Rare
Autoimmune Cholangitis	5% to 10% of PBC patients		Chronic hepatic inflammation; variant syndrome of autoimmune hepatitis	Fatigue, pruritus, cholestasis, bile duct injury followed by ductopenia with little or no portal inflammation	Unknown
Biliary Atresia	1:10,000–15,000 (50% of pediatric liver transplants)		80% to 90% perinatal form: unknown etiology, probably due to prenatal or perinatal viral infection; bile duct injury, inflammation, and obstructive fibrosis. 10% to 20% prenatal (or early severe) form: Likely genetic	Jaundice and alcoholic stools due to fibro-obliterative obstruction of the bile ducts. Frequent progression in secondary biliary cirrhosis with splenomegaly and portal hypertension	Unknown
Caroli’s disease	1:1,000,000	PKHD1 (fibrocystin/polyductin)	Ductal plate malformation leading to necroinflammation of the biliary epithelia	Recurrent cholangitis, biliary stones and cyst complications	6% to 30% of cases
Cystic Fibrosis	1:3000	CFTR (cAMP-dependent Cl-channel)	Abnormal chloride conductance on the apical membrane of the epithelial cells	Neonatal cholestasis, liver steatosis, hepatomegaly, focal biliary cirrhosis, and liver cirrhosis with or without portal hypertension.	Unknown
Primary Biliary Cholangitis (PBC)	1:2500		Reduced expression of the bicarbonate transporter (SLC4A2) on the apical cholangiocyte domain. Non-suppurative inflammation and destruction of the interlobular bile ducts	Cholestasis, biliary cirrhosis, serum antimitochondrial antibodies, end-stage liver disease	Unknown
Primary Sclerosing Cholangitis (PSC)	0–16.2:100,000		Chronic inflammation in bile ducts, immune-mediated association with inflammatory bowel disease	Chronic inflammation of intrahepatic and extrahepatic bile ducts, with obliterative cholangitis and progression to cirrhosis	10% of cases

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
