# Peer review of "Fibroinflammatory Liver Injuries as Preneoplastic Condition in Cholangiopathies"

_ijms, 2018, doi:10.3390/ijms19123875_

Reviewer 1 Report
This is a very interesting and well written review on fibroinflammatory liver injuries as preneoplastic condition in cholangiopathies. The literature is appropriate and updated. The structure of the manuscript is reasonable and clear. In general, this is an excellent manuscript. I only have a minor suggestion. The authors have mentioned the role of toxic insults in the development of cholangiocarcinoma, for example following chronic oral administration of thioacetamide (TAA). They may like to distinguish between xenobiotics and endogenous compounds and between carcinogenic and cocarcinogenic agents. At this respect, it would be interesting to mention the co-carcinogenic effects of intrahepatic bile acid accumulation in cholangiocarcinoma development (PMID: 24255171).
Author Response
We would like to thank the reviewer for the positive remark on our manuscript. Our answer is enclosed below.
“They may like to distinguish between xenobiotics and endogenous compounds and between carcinogenic and cocarcinogenic agents. At this respect, it would be interesting to mention the co-carcinogenic effects of intrahepatic bile acid accumulation in cholangiocarcinoma development (PMID: 24255171).” Thank you for this suggestion; statement regarding the pro-neoplastic effects of endogenous compounds as well as of xenobiotic carcinogens and cocarcinogens were added at page 11-12. Moreover, with regards to these arguments, two new reference were added to the paper, ref. number 124 and 125.
Reviewer 2 Report
The review by Cannito et al. describes the complex interplay between inflammation and fibrosis in the context of cholangiopathies with how this process contributes to the development of CCA. The authors describe relevant studies and do a good job with describing the literature. A significant strength of this review is delineating the crosstalk and signaling between cholangiocytes, hepatic stellate cells and immune cells. Therefore, this review is important and relevant to the field. That being said, I have a few minor items that should be corrected by the reviewers:
1) Primary biliary cirrhosis should be changed to primary biliary cholangitis.
2) Line 146: is the 27 on this line a reference?
3) Line 279-280: TGF beta 1 being released from the LAP/LTBP complex would not be considered gain of function in my opinion. It would be best to say that TGFB1 is released from this protein complex as a monomer and will dimerize with itself giving it the ability to bind TGFbeta receptors.
4) Line 362: A word is missing after “and”.
5) Minor wording errors throughout. An example is line 273 “Despite de fact”. Please read and correct these throughout.
Author Response
We would like to thank the reviewer for the positive considerations and the constructive comments on this review. Answers to the questions raised appear below.
1) “Primary biliary cirrhosis should be changed to primary biliary cholangitis.” In accordance with the new guidelines proposed and with the punctual recommendation of the Reviewer, we amended the text as suggested-
2) “Line 146: is the 27 on this line a reference?”. We apologize for this mistake: 27 is referred to a reference. We amended this error and we changed the other references accordingly.
3) “Line 279-280: TGF beta 1 being released from the LAP/LTBP complex would not be considered gain of function in my opinion. It would be best to say that TGFB1 is released from this protein complex as a monomer and will dimerize with itself giving it the ability to bind TGFbeta receptors”. We agree with this consideration of the Reviewer and we rephrased the sentence as recommended. Changes were made at page 8, line 282-283. We would like to thanks again the referee for the suggestion.
4) “Line 362: A word is missing after “and””. We apologize for this oversight; the missing phrase “chemokine (C-X-C motif) ligand” was added to page 9, line 366.
5) “Minor wording errors throughout. An example is line 273 “Despite de fact”. Please read and correct these throughout”. We are very sorry for the typos and errors along the text. We have thoroughly revised the manuscript, and correct the mistakes along the text.